# Oxidative Stress and DNA Damage in Peripheral Blood Mononuclear Cells from Normal, Obese, Prediabetic and Diabetic Persons Exposed to Thyroid Hormone In Vitro

**DOI:** 10.3390/ijms23169072

**Published:** 2022-08-13

**Authors:** Ninoslav Djelić, Sunčica Borozan, Vesna Dimitrijević-Srećković, Nevena Pajović, Milorad Mirilović, Helga Stopper, Zoran Stanimirović

**Affiliations:** 1Department of Biology, Faculty of Veterinary Medicine, University of Belgrade, 11000 Belgrade, Serbia; 2Department of Chemistry, Faculty of Veterinary Medicine, University of Belgrade, 11000 Belgrade, Serbia; 3Clinic for Endocrinology, Diabetes and Metabolic Diseases, Clinical Center of Serbia, 11000 Belgrade, Serbia; 4Department of Economics and Statistics, Faculty of Veterinary Medicine, University of Belgrade, 11000 Belgrade, Serbia; 5Institute of Pharmacology and Toxicology, University of Würzburg, 97070 Würzburg, Germany

**Keywords:** diabetes, oxidative stress, DNA damage, lymphocytes, thyroid hormone

## Abstract

Diabetes, a chronic group of medical disorders characterized byhyperglycemia, has become a global pandemic. Some hormones may influence the course and outcome of diabetes, especially if they potentiate the formation of reactive oxygen species (ROS). There is a close relationship between thyroid disorders and diabetes. The main objective of this investigation was to find out whether peripheral blood mononuclear cells (PBMCs) are more prone to DNA damage by triiodothyronine (T_3_) (0.1, 1 and 10 μM) at various stages of progression through diabetes (obese, prediabetics, and type 2 diabetes mellitus—T2DM persons). In addition, some biochemical parameters of oxidative stress (catalase-CAT, thiobarbituric acid reactive substances—TBARS) and lactate dehydrogenase (LDH) were evaluated. PBMCs from prediabetic and diabetic patients exhibited increased sensitivity for T_3_ regarding elevated level of DNA damage, inhibition of catalase, and increase of TBARS and LDH. PBMCs from obese patients reacted in the same manner, except for DNA damage. The results of this study should contribute to a better understanding of the role of thyroid hormones in the progression of T2DM.

## 1. Introduction

In the last few decades, there was a significant increase of newly discovered cases of diabetes, making it a global pandemic. Diabetes mellitus is a group of metabolic diseases, causing hyperglycemia due to decreased insulin secretion and/or lack of its adequate effects on insulin receptors. The most prevalent form of diabetes is type 2 diabetes mellitus (T2DM) [1], and it is expected to increase to 693 million cases worldwide by the year 2045 [2]. It is noteworthy that one of the most important factors causing this disease is obesity [3].

There is increasing evidence that T2DM is associated with oxidative stress [4]. Namely, an oxidative stress condition is elevated in T2DM patients, and it is accompanied by a significant increase in biochemical parameters of oxidative stress in comparison to healthy individuals [5]. Oxidative stress causes damage of biomacromolecules including DNA [5]. T2DM usually progresses slowly, and even patients with prediabetes already exhibit an increase in oxidative stress levels [6]. In the cells of diabetic patients the amount of glucose oxidized in the Krebs tricarrboxylic acid cycle is increased, thus resulting in an elevated production of reactive oxygen species (ROS). There is evidence that oxidative stress plays a key role in the pathogenesis and progression of diabetes and its complications, such as foot ulcers, diabetic neuropathy and nephropathy, myocardial infarction, and cerebrovascular accidents [7].

Thyroid hormone (TH, triiodothyronine, T_3_) accelerates oxidative metabolism and production of ROS, and by doing so it is capable of worsening diabetes. TH increases ROS levels by accelerating basal metabolism, stimulating the synthesis of respiratory chain components, and influencing the expression of genes encoding enzymes involved in the formation and elimination of ROS [8].

There is a complex interplay between thyroid disease and diabetes [9]. For example, insulin resistance is present in both hypo- and hyperthyroidism [10]. Elevated levels of thyroid hormone cause an increase in glucose output from liver, which causes hyperinsulinemia, induction of glucose intolerance, and development of peripheral insulin resistance [11].

Based on the above, we hypothesize that PBMCs of prediabetic and diabetic individuals may be more sensitive to the DNA damaging effect of adrenaline and oxidative stress, which may result in worsening diabetes control. In order to investigate our hypothese, we determined the genotoxic effects and as parameters of oxidative stress catalase (CAT), thiobarbituric acid reactive substances (TBARS), and lactate dehydrogenase (LDH) after administration of adrenaline in cells from healthy, obese, prediabetic, and diabetic persons. To our knowledge, this type of study has not yet been conducted, and the observed results may contribute to better understanding the significance of adrenaline in progression of T2DM.

Based on this, we made a hypothesis that PBMCs in prediabetic and diabetic persons may be more susceptible to creation of DNA damage and oxidative stress, following exposure to T_3_, in comparison to the PBMCs of normal persons. Thus, we decided to investigate the sensitivity of peripheral blood mononuclear cells (PBMC) to T_3_ from persons in various phases of progression through T2DM (normal, obese, prediabetics, and persons with T2DM) regarding oxidative stress markers and DNA damage. To investigate this, we analyzed the genotoxic effects in the Comet assay and parameters of oxidative stress (catalase—CAT, thiobarbituric acid reactive substances—TBARS) and lactate dehydrogenase (LDH) after in vitro administration of T_3_ to PBMCs of normal, obese, prediabetic, and diabetic persons. To our knowledge, this kind of investigation was not yet conducted, and the obtained results may contribute to a better understanding of a possible role of TH in the progression of T2DM.

## 2. Results

### 2.1. Analysis of DNA Damage in the Comet Assay

The results of total comet score (TCS) values as an indicator of primary DNA damage in various groups (healthy, obese, prediabetics, and T2DM patients) are presented in Figure 1. In all groups of patients only the lowest concentration of T_3_ (0.1 μM) did not increase the value of TCS, while concentrations of 1 and 10 μM caused a significant increase in primary DNA damage. Therefore, the reactivity of PBMCs to T_3_ was very similar at all groups representing progression through various stages of T2DM. As for the positive control (100 μM hydrogen peroxide, H_2_O_2_), it exhibited the most profound DNA-damaging effects of all four analyzed groups.

### 2.2. Antioxidant Defence Enzyme Activity

The results of CAT activity are presented in Figure 2. It can be noted that T_3_ caused a slight increase of CAT activity in obese subjects with the highest dose (*p* < 0.05). However, there was a statistically significant decrease of the elevated CAT activity in PBMCs from prediabetics and T2DM patients after treatment with all tested concentrations of T_3_ (0.1, 1 and 10 μM) in comparison to the negative control. Namely, the activity of the enzyme was lowered by 15.49% (*p* < 0.05), 23.35% (*p* < 0.001), and 33.38% (*p* < 0.0001) in prediabetics and 12.76%, 28.69%, and 29.78% (*p* < 0.01) in diabetic patients. A similar percentage of CAT inhibition was caused by H_2_O_2_ in both groups of subjects, 32.20% and 34.31% (*p* < 0.001).

### 2.3. Degree of Cell Membrane Damage

In order to examine the influence of increased ROS production on the level of cell membrane damage in the lymphocytes of the examined subjects (normal, obese, prediabetics, and T2DM), under the influence of various concentrations of T_3_ in vitro, we determined the concentrations of MDA via TBARS, as well as activity of LDH as a specific marker of cell membrane damage in lymphocytes. The results of the effects of various concentrations of T_3_ on cell membrane damage are presented in Figure 3.

In the lymphocytes of healthy persons there was a slight increase in the cell membrane damage at all tested concentrations, but it did not reach statistical significance in comparison to the negative control level. In lymphocytes of obese subjects there was a significant increase of membrane lipid damage at all concentrations of T_3_: 0.1, 1, and 10 μM (*p* ˂ 0.05, *p* < 0.01 and *p* < 0.001). The concentration of MDA was increased in comparison to the untreated lymphocytes for 23.13%, 29.85%, and 40.41% for increasing concentrations of T_3_ of 0.1, 1, and 10 μM, respectively. The examined concentrations of T_3_ caused an increase of lipid peroxidation in prediabetics with the level of significance of *p* < 0.05 (0.1 μM T_3_) and *p* < 0.01 (1 and 10 μM T_3_). The concentration of MDA was increased in comparison to the untreated lymphocytes for 57.96%, 70.71%, and 95.65% for increasing concentrations of T_3_ of 0.1, 1, and 10 μM, respectively. However, in T2DM subjects all concentrations of T_3_ caused an obvious damage of membrane lipids at the level of statistical significance of *p* ˂ 0.0001. The concentration of MDA was increased in comparison to the untreated lymphocytes for 24.16%, 46.14%, and 70.94% for increasing concentrations of T_3_ of 0.1, 1, and 10 μM, respectively. In summary, significant differences between the investigated groups with respect to harmful effects of T_3_ on the cell membrane were most obvious in prediabetic and T2DM patients. Lymphocytes treated with H_2_O_2_ also had cell membrane damage, and the level of statistical significance was different: *p* < 0.01 in obese and prediabetics, and *p* < 0.0001 in T2DM patients.

### 2.4. Analysis of Extracellular LDH

The results of the amount of released enzyme LDH in all investigated groups are presented in Figure 4.

On the basis of the amount of this enzyme, we can conclude that there is a tendency of release of LDH in all groups treated with various doses of T_3_ with a high level of statistical significance. The most intensive changes are obtained in prediabetics and T2DM groups treated with concentrations of 0.1 (*p* < 0.01) and 1 μM of T_3_ (*p* < 0.001), with a maximal expression at 10 µM of T_3_ (*p* < 0.0001). The same effect was observed with H_2_O_2_ (*p* < 0.0001).

### 2.5. Analysis of Area under the Curve

Results of the analysis of the area under the curve (AUC) for each investigated parameter are presented in Figure 5. On the basis of the obtained results, it is obvious that the total comet score (TCS) increased (*p* < 0.0001) in all investigated groups (obese, prediabetics and diabetics), when compared to normal subjects. The values of AUC for CAT activity, TBARS, and LDH revealed that only the group of obese patients did not differ significantly from normal persons (control group). However, prediabetics and T2DM groups had a decreased level of CAT activity and increased levels of TBARS and LDH in comparison to normal and obese persons, with various levels of statistical significance (Figure 5).

## 3. Discussion

Increased oxidative stress has a central role in the pathogenesis of many diseases, such as atherosclerosis, vascular inflammation, and endothelial dysfunction, and it is thought to play a role in the progression of prediabetes to diabetes [12]. Thus, it is important to detect changes in oxidative stress early enough to prevent disease progression. There are numerous scientific reports that an increased production of (ROS) in patients with diabetes contributes to disease progression and chronic complications [13,14]. Patients with diabetes have both an increased level of oxidative DNA damage and lower capacity of antioxidative defence mechanisms [15].

In this investigation, we treated human peripheral blood mononuclear cells with three concentrations of T_3_ (0.1, 1 and 10 μM). In PBMCs of all groups of blood donors (normal, obese, prediabetes and T2DM), the concentration of 0.1 μM did not significantly change DNA damage in comparison to the negative control. In contrast, the remaining higher concentrations (1 and 10 μM of T_3_) elevated DNA migration in all investigated groups. Therefore, the pattern of DNA damage in the Comet assay is similar in various phases of progression through the T2DM. However, the area under the curve (AUC) analysis showed that obese, prediabetics, and diabetics when compared to the control group had a significantly higher response in the Comet assay evaluated using the total comet score (TCS) parameter (Figure 5). Therefore, even obesity poses an increased risk of DNA damage due to an increased sensitivity of PMBCs to the genotoxic effects of T_3_. These results are consistent with previous reports of increased oxidative DNA damage in prediabetic subjects, and even greater damage in T2DM subjects [9,16,17]. Hence, it is not surprising that obesity contributes to the ethiopathogenesis of diabetes complications [18]. Moreover, results of this investigation are in accordance with our previous finding that PMBCs from obese, prediabetics, and diabetics treated with the stress hormone adrenaline had a higher level of DNA damage in comparison to normal persons [19].

Catalase is a very common enzyme, which is found in nearly all living organisms exposed to oxygen. The enzyme catalyzes the decomposition of H_2_O_2_ into oxygen and water. Moreover, it plays a vital role in protecting the cell from oxidative damage which is caused by ROS. The H_2_O_2_ concentrations can be increased through the acquired deficiencies in the levels of the enzyme. It can create both a physiological and a toxic effect. However, the increased concentrations of H_2_O_2_ are a risk factor for the patients with T2DM [20].

In this study, we showed that the treatment of lymphocytes from patients with prediabetes and T2DM with various concentrations of T_3_ or H_2_O_2_ leads to a decreased activity of CAT. Therefore, the antioxidative capacity of the cell is lowered and production of free radicals is increased, and all of that results in redox homeostasis disorder [21]. ROS can inactivate CAT and cause numerous harmful effects due to accumulation of H_2_O_2_ and superoxide anion [22]. Relatively high concentrations of H_2_O_2_ are released at the site of any kind of inflammation, but also in uninflammed cells (fibroblasts, endothelial cells, chondrocytes) under the influence of cytokines, such as interleukin-1 and TNF-α. Cells are protected from H_2_O_2_ by powerful enzyme systems, such as CAT in peroxisomes and glutathione peroxidase in mitochondrion [23]. Catalase is an enzyme with a relatively high expression in almost all tissues. Early and profound oxidative stress in any organ, especially in the pancreas, leads to a probably compensatory increase in the activity of antioxidative enzymes, including CAT [24].

If the H_2_O_2_ is not removed by CAT it can react with ferric ions (the Fenton reaction), making the more potent hydroxyl radical. This radical causes mitochondrial membrane damage in situ leading to loss of ATP production, damage to DNA, cell membrane by modification of membrane proteins, lipid peroxidation, and possibly irreversible cell destruction [20].

There is increasing evidence that oxidative stress is involved in pathogenesis of T2DM. On the other hand, hyperglycemia, as one of the hall-marks of diabetes, influences early stages of T2DM development. For example, erythrocytes from blood are exposed to an increased glucose, which can affect the integrity of their cell membranes, production of ROS, changes in hexosomonophospate shunt, glutathione cycle, and antioxidative defence. An increased glucose level also influences hemoglobin modification, reflecting on the Bp3 function [25,26].

Oxidative damage caused by ROS also includes lipid peroxidation. The end products of lipid peroxidation are very reactive and may cause consecutive chain propagation and oxidation of nearby polyunsaturated fatty acids (PUFA). Products of lipid peroxidation can be transported from the site of their formation to the other sites in organism transfering an information about the origin of primary oxydative damage [24,27]. In this investigation, the extent of lipid peroxidation was evaluated by concentration of TBARS, including the MDA. MDA is one of the end products of products of cyclic reaction of lipid peroxidation. It has been shown that MDA exhibits mutagenic and carcinogenic effects in mammalian cells [28,29]. Increased levels of TBARS were not detected for all three concentrations of T_3_ in the PBMCs from the normal subjects (control group), when compared with the negative control. However, in PBMCs from obese, prediabetics, and T2DM patients there was an increase of TBARS at all three concentrations of T_3_ in vitro in comparison to the negative control (Figure 3). There are some experimental findings that increased TBARS in patients with T2DM are in direct correlation with the duration of the disease and development of complications [30,31].

Thyroid hormones influence lipid composition of rat tissues and consequently the susceptibility to oxidative stress. However, the response is tissue-specific, and discrepant effects of T_3_ and T_4_ have been reported. In rat liver, T_3_ hyperthyroidism was found to be associated with altered lipid-peroxidation indexes, including elevated levels of TBARS and lipid hydroperoxides that are byproducts of lipid peroxidation [32,33].

In order to indirectly monitor the level of lipid peroxidation, we also evaluated release of the enzyme LDH from cells. Namely, after the damage of cell membranes there is a leak of cytoplasmic LDH from cells. The more enzyme of LDH is released, the greater the damage of cell membranes [34]. In this investigation, the amount of released LDH was higher in persons with prediabetes and T2DM. The obtained results of two different parameters of lipid peroxidation (MDA and LDH) are elevated in the same groups of patients (prediabetics and T2DM), confirming the reliability in determination of membrane damage.

Despite the results obtained in this study, one of the limitations is that we did not analyze concentrations of ROS after the exposure of PBMCs to T_3_. Since the half-life of ROS is very short, we preferred to analyze more stable parameters—antioxidative enzymes and damage of the cell membranes instead.

Moreover, ROS can be closely linked with progression of inflammatory disorders and tissue injury. Although deleterious to cells at high enough concentrations, at low concentrations ROS can serve as signaling molecules capable of regulating cell growth, differentiation, cell senescence, and apoptosis. Chronic or prolonged ROS production has an important role in the progression of inflammatory disorders. ROS produced by mitochondrion are implicated in chronic inflammation, cancer progression, diabetes mellitus, and atherosclerosis, and it also contributes to LPS-mediated production of pro-inflammatory cytokines IL-1β, IL-6, and TNF-α [35]. These data point to the relationships of ROS production, inflammation, and damage of biomolecules. In our study, all patients were overweight (25 < BMI < 30), or obese (BMI > 30). Increased waist circumference is characterised by inflammation of lipid tissue and decreased antioxidative defence, leading to increase of ROS production, especially in T2DM patients [36].

The generation of ROS starts the lipid peroxidation interacting with PUFA, the concentration of which is especially high in the cell membranes of parenchymatic organs in T2DM patients [37]. Thus, it is very important to monitor parameters of oxidative stress, such as H_2_O_2,_ in order to provide an answer about formation of highly reactive hydroxyl radical capable to induce more damage favoring the processes of mutagenesis and carcinogenesis [38]. Despite some progress in understanding the role of TH in the processes of mutagenesis, further investigation is needed in order to unravel precise molecular mechanisms of T_3_ induced genotoxicity.

## 4. Material and Methods

### 4.1. Study Subjects

The present study was conducted on 28 females from 46 to 63 years old (7 persons in each group with the following mean ± SEM age: healthy 53.0 ± 3.6; obese 61.0 ± 1.4; prediabetes 53.8 ± 2.3; T2DM 57.0 ± 2.9), who attended the Department of Endocrinology, Diabetes and Metabolic Diseases in Belgrade (Table 1). Neither of the subjects had been ever diagnosed with malignancy, heart infarction, cerebrovascular insult, or any other serious complication of diabetes. All of them were non-smokers, under no medication (except for metformin for controlling blood sugar level), and without use of any food supplements. All participants were informed about the procedure of the study and signed a written consent, which was approved by the Ethical committee at the Faculty of Pharmacy (University of Belgrade, Approval No. 1103/2). The study was conducted according to the Guidelines of the Declaration of Helsinki, and all participates gave written consent and authorization to publish this paper.

Participants were divided into four groups (seven females in each group): female individuals with normal blood glucose levels (healthy group); group with normal blood glucose levels and body mass index (BMI) ≥ 30.0 kg/m^2^ (obese group); group with impaired fasting blood glucose (5.5 mmol/L < FBG < 7 mmol/L) (prediabetic group)—according to the recommendations of the World Health Definition, Diagnosis and Classification of Diabetes Mellitus and its Complications [39], and group selected on the basis of having been previously diagnosed with T2DM using an oral glucose tolerance test and/or being on antihyperglycaemic medication (diabetic group) (Table 1).

The results are presented as mean ± SE; FPGfasting plasma glucose; BM—body mass; BH—body height; BMI—body mass index; and BP- blood pressure. There were no statistically significant differences among groups for any analyzed parameter (*p* > 0.05 in one-way ANOVA and Tukey’s test).

### 4.2. Sampling and Cell Preparation

Blood samples were withdrawn by venipuncture from female donors and blood was diluted (1:1) with RPMI medium, underlaid with Histopaque, and centrifuged at 1900× *g* for 15 min. The lymphocyte layer (buffy coat) was washed twice in RPMI 1640 medium, each wash was followed by centrifugation for 10 min at 1800× *g*.

### 4.3. Comet Assay

The alkaline version of Comet assay was performed according to Singh et al. [40] and Tice et al. [41], with minor modifications. Briefly, a suspension of isolated lymphocytes in PBS was treated with TH (0.1, 1, and 10 μM, CAS No. 51-43-4) for 30 min at 37 °C. Each experiment included a positive control (100 μM H_2_O_2_) and negative control (untreated cells). After incubation, 100 μL of cell suspension was mixed with 100 μL of 1% low melting point agarose (LMPA). A 90 μL suspension was casted on a microscope slide precoated with 1% normal melting agarose and put in the fridge to solidify. Then, the slides were immersed in cold lysis solution at pH 10 (2.5 M NaCl, 100 mM EDTA, 10 mM Tris pH 10, 1% Triton X–100, 10% DMSO) overnight at 4 °C. After lysis, slides were placed in a horizontal gel electrophoresis tank to allow DNA unwinding in cold electrophoresis buffer (300 mM NaOH, 1 mM EDTA, pH > 13) for 30 min. Electrophoresis was carried out at 4 °C with an electric current of 25 V (1.1 V/cm) for 30 min. The slides were then removed from the tray and washed with a neutralizing buffer (0.4 M Tris HCl, pH 7.5) for 5 min. The neutralization was repeated three times. Finally, the slides were fixed with ice cold methanol, dried, and stored. Before analysis, the slides were rehydrated with ice cold distilled water and stained with ethidium bromide (20 μg/mL). The slides were examined under a fluorescence microscope (AxioImager Z1, Carl Zeiss; excitation filter, 515–560 nm; emission filter, 590 nm). Comet were scored visually as described by Anderson et al. [42] and expressed as the (TCS), according to Collins [43]. In visual scoring, the estimated level of DNA damage results in five classes of comets: (A) no damage, <5%; (B) low level damage, 5–20%; (C) medium level damage, 20–40%; (D) high level damage, 40–95%; and (E) total damage, >95% (“hedgehogs comets”). (Figure 1). TCS is calculated according to the formula:TCS = 2 × B + 3 × C + 4 × D + 5 × E
where B to E represents percentage of cells with various level of DNA damage.

### 4.4. Determination of Catalase (CAT)

CAT activity was assayed using H_2_O_2_ as a substrate and activity was followed by measuring the decrease in absorbance at 240 nm [44]. The enzyme activity was expressed in U/mL.

### 4.5. Determination of Lipid Peroxidation (MDA)

Lipid peroxidation was assessed in suspension of isolated PBMCs in PBS after treated with T_3_, by measuring (TBARS) test according to the methods Gutteridge [45] and Traverso et al. [46]. The assay evaluated the formation of a colored adduct after the stoichiometric reaction between thiobarbituric acid (TBA) and several lipid derived aldehydes, including (MDA). The TBARS level released in the samples was measured at 535 nm. The results were expressed in nmol/mL. Each experiment included a positive control (100 μM H_2_O_2_) and a negative control (untreated cells).

### 4.6. Determination of Lactate Dehydrogenase (LDH)

Extracellular total (LDH) was determined after the incubation period, suspension PBMCs in PBS with T_3_, centrifugation for 5 min at 2000 rpm and in supernatant determines LDH by the method of Bergmeyer and Brent (1974) [47]. In the reaction between sodium pyruvate and NADH at pH 7.2, lactate is formed and reacts with the enzyme from the sample. The activity of LDH is determined by the decrease in absorbance at 340 nm and expressed as enzyme activity per liter supernatant (U/L). All spectrophotometric measurements were done using a Cecil CE 2021 UV/VIS spectrophotometer (Cecil Instruments Ltd., Cambridge, UK).

### 4.7. Statistical Analysis

Due to the homogeneity of the data obtained in all measurements (coefficient of variation <30%), one-way analysis of variance (ANOVA) was used, followed by a Dunnett test. One way ANOVA was used to compare AUC values, and subsequent comparisons were made with the Tukey post-hoc test. All values are presented as the mean ± SE. A *p* value of less than 0.05 was considered to be statisticaly significant. Statistical analysis was done using GraphPad Prism 6.0 Software, San Diego, CA, USA.

## 5. Conclusions

PBMCs of obese, prediabetic, and T2DM patients are more susceptible to DNA damage in the Comet assay after in vitro treatment with T_3_ in comparison to those of healthy individuals. Additionally, in these groups of patients, treatment with T_3_ caused an increase in oxidative stress, damage of the cell membrane, and decrease in enzymatic antioxidative defence in prediabetics and T2DM patients in comparison to healthy subjects. The results of this study should contribute to a better understanding of the role of thyroid hormones in the progression of T2DM.

## Figures and Tables

**Figure 1 ijms-23-09072-f001:**
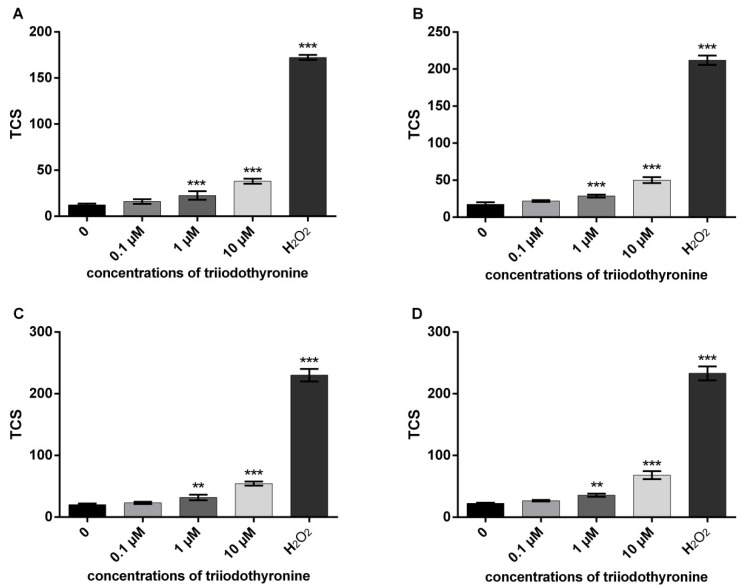
DNA damage in various stages of progression through diabetes mellitus type 2: (**A**)—normal persons; (**B**)—obese with normal glycoregulation; (**C**)—patients with prediabetes; (**D**)—patients with type 2 diabetes mellitus (T2DM). TCS—total comet score; ** *p* < 0.01; *** *p* < 0.001—comparisons with the negative control (0 point), in one-way ANOVA by Dunnett test.

**Figure 2 ijms-23-09072-f002:**
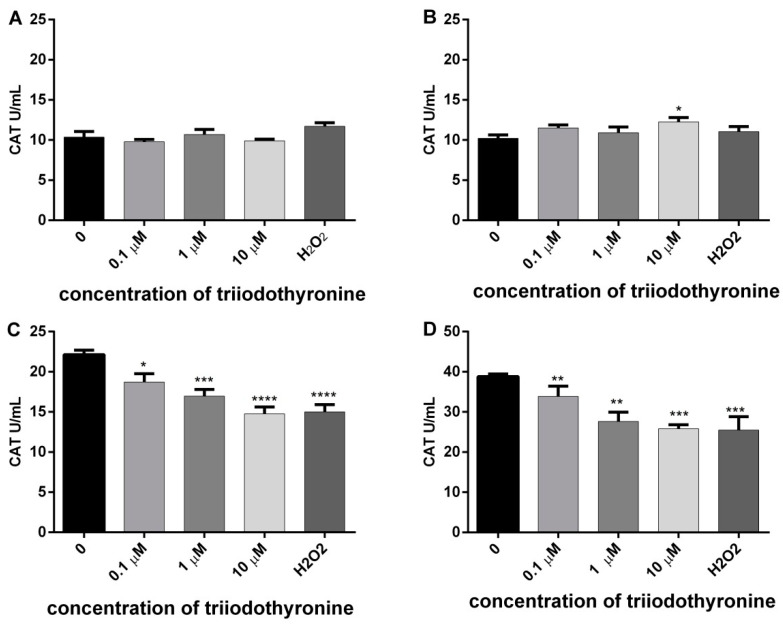
Activity of catalase (CAT) in lymphocytes of (**A**) normal, (**B**) obese, (**C**) prediabetics, and (**D**) DMT2 persons treated with vireos concentrations of triiodothyronine and (T_3_) positive control (H_2_O_2_). The results are presented as means ± standard error of mean (SEM). * *p* < 0.05, ** *p* < 0.01; *** *p* < 0.001; **** *p* < 0.0001 in comparison to the negative control, in one-way ANOVA by Dunnett test.

**Figure 3 ijms-23-09072-f003:**
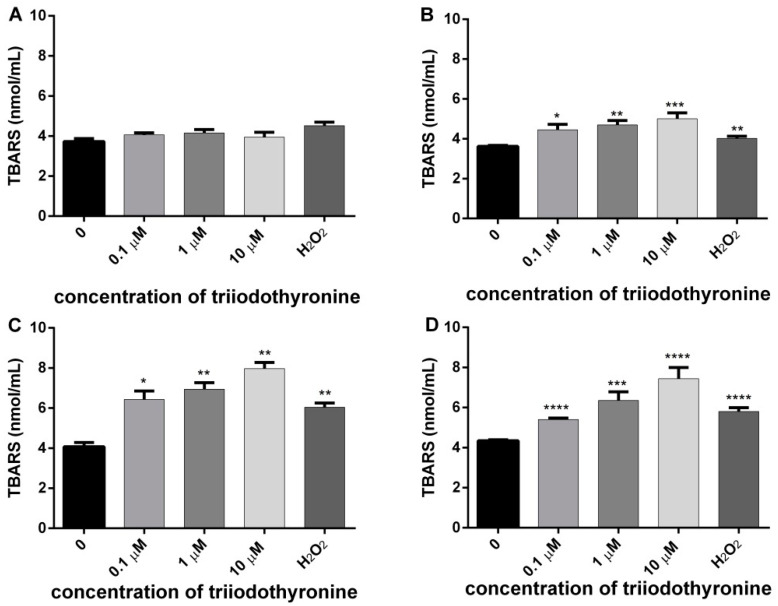
Concentrations of malondialdehyde (MDA) in lymphocytes of (**A**) normal, (**B**) obese, (**C**) prediabetics, and (**D**) T2DM patients treated with various concentrations of triiodothyronine and the positive control (H_2_O_2_). The results are presented as means ± SEM. * *p* < 0.05, ** *p* < 0.01; *** *p* < 0.001; **** *p* < 0.0001 in comparison to the negative control, in one-way ANOVA by Dunnett test.

**Figure 4 ijms-23-09072-f004:**
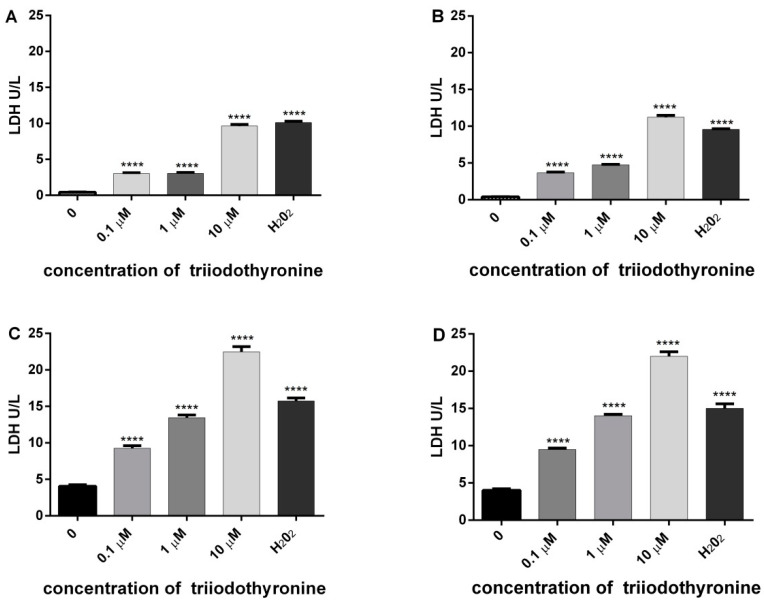
Level of lactate dehydrogenase (LDH) release as an indicator of cell cytotoxicity in lymphocytes of (**A**) normal, (**B**) obese, (**C**) prediabetics, and (**D**) T_2_DM, treated with various concentrations of triiodothyronine and positive control (H_2_O_2_). The results are presented as means ± SEM. **** *p* < 0.0001 in comparison to the negative control group, in one-way ANOVA by Dunnett test.

**Figure 5 ijms-23-09072-f005:**
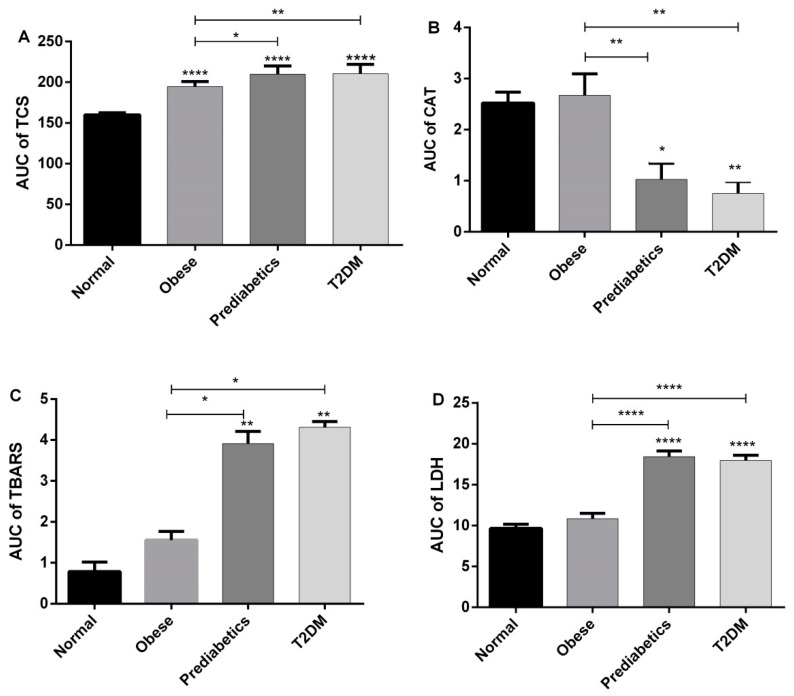
Results of the analysis of area under the curve (AUC)—comparisons among groups for each of investigated parameters: (**A**) TCS in the Comet assay; (**B**) catalase; (**C**) TBARS, and (**D**) LDH * *p* < 0.05; ** *p* < 0.01, **** *p* < 0.0001, in one-way ANOVA and Tukey’s test.

**Table 1 ijms-23-09072-t001:** Basic characteristics of the study groups.

Parameters	Normal	Obese	Prediabetics	T2DM
Age (years)	53.00 ± 3.54	61.00 ± 1.41	53.83 ± 2.27	57.00 ± 2.94
FPG (mmol/L)	5.16 ± 0.30	5.30 ± 0.22	5.57 ± 0.17	5.45 ± 0.33
HbA1c (%)	5.68 ± 0.16	6.05 ± 0.29	5.70 ± 0.11	6.05 ± 0.19
BM (kg)	70.78 ± 9.72	84.65 ± 2.96	85.30 ± 8.77	78.75 ± 6.65
BH (cm)	165.20 ± 2.52	167.00 ± 2.74	165.67 ± 4.12	175.50 ± 3.84
BMI (kg/m^2^)	26.16 ± 3.46	30.40 ± 0.27	30.77 ± 2.40	25.40 ± 1.27
Systolic BP (mmHg)	124.00 ± 4.00	132.50 ± 10.31	121.67 ± 9.10	122.50 ± 2.50
Diastolic BP (mmHg)	82.00 ± 2.00	82.50 ± 4.79	76.67 ± 5.58	80.00 ± 0.00

## Data Availability

The data that supports the findings of this study are available from the corresponding author upon reasonable request.

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
