# Peer review of "Oxidative Stress and DNA Damage in Peripheral Blood Mononuclear Cells from Normal, Obese, Prediabetic and Diabetic Persons Exposed to Thyroid Hormone In Vitro"

_ijms, 2022, doi:10.3390/ijms23169072_

Round 1
Reviewer 1 Report
Thank you for the opportunity to review your article. It is a very well-prepared, with a well-designed methodology and illustrative results. I am only making a few minor suggestions and comments below.
In the article you mention that patients receiving antihyperglycaemic medication (diabetic group) are included. Please define whether only those patients receiving insulin or also those receiving tablets are meant here?
For the methodology, please define in more detail how you have included the participants, whether they have signed any authorisation. Also describe how long you will keep the data, who has access to it.
Expand on the conclusion and link it more closely to the findings.
Add the limitations of the research.
Standardise references and follow the guidelines for references.
Author Response
First of all, on behalf of all co-authors I would like to thank you for your time and suggestions which make this manusrcipt much better. We have accepted all suggestions, and all changes in the revised version of the manuscript are marked in red.
In the article you mention that patients receiving antihyperglycaemic medication (diabetic group) are included. Please define whether only those patients receiving insulin or also those receiving tablets are meant here?
Response: Patients with prediabetes and diabetes were on metformin medication; at pages 15-16, section 4.1. Study subjects, we inserted a text: “(except for metformin for controlling blood sugar level)”.
For the methodology, please define in more detail how you have included the participants, whether they have signed any authorisation.
Response: All patients were informed about study in detail, and have given a written consent to use their blood only for this purpose. In the first paragraph at the page 16 we added a word “authorisation”.
Also describe how long you will keep the data, who has access to it.
Response: Data are only available to the reserachers in this study.
Expand on the conclusion and link it more closely to the findings.
Response: Thank you very much for this suggestion, we expanded the Conclusions, at pages 19-20 you can find inserted text marked in red.
Add the limitations of the research.
Response: At page 14 we inserted the following text: “Despite the results obtained in this study, one of the limitations is that we did not analyse concentrations of ROS after the exposure of PBMCs to T3. Since the halflife of ROS is very short, we prefered to analyse more stable parameters - antioxidative enzymes and damage of the cell membranes instead.”
Standardise references and follow the guidelines for references.
Response: Thank you for this suggestion, we standardised all the references according to the guidelines of the journal.
Reviewer 2 Report
In this manuscript, the sensitivity of peripheral blood mononuclear cells (PBMC) to T3 in normal, obese, prediabetics and persons with T2DM, regarding oxidative stress markers and DNA damage was investigated. The scientific work is very interesting, however, some aspects, as indicated below, should be addressed before the document can be considered for publication in this journal. This version of the manuscript is not enough complete.
Minor revision:
· English language and style are fine/minor spell check required.
· I suggest to review the style of the manuscript according to the guidelines of the journal.
· All acronyms in the Abstract and Manuscript should be defined when first appear.
Major revision:
· There is an increasing evidence that T2DM is associated with oxidative stress. Oxidative stress is critically involved in diabetes pathogenesis and hyperglycemia is the principal factor in the early stages of development. In Blood, erythrocytes are mainly exposed to glucose and other molecules present in the blood (i.e. oxidant compounds), and they have been widely investigated for their important role in different physiological conditions due to the fact of their metabolism and sensitivity to oxidative stress. Thus, the authors should add information about the link between oxidative stress increase and red blood cells. I suggest some recent papers (10.3390/antiox9050365) (10.1371/journal.pone.0235335), in order to provide a general more knowledge.
· Aim should be explain better.
· In Figure 1, total comet score, * p <0.05; ** p <0.01; *** p <0.001 (Bonferroni test) -comparisons with the negative control (0 point)- is showed. In the caption, a double comparison is missing. However, the author indicate the following test Bonferroni test. Statistical analyzes should be explain better. Maybe by Dunnett test.
· The authors investigated about the assessment of oxidative stress in peripheral blood mononuclear cells. However, ROS levels are missing. Why? It could be a direct evidence when oxidative stress is produced in excess.
· Oxidative stress is often associated to inflammation. No evidence are showed about the link between ROS production and targets of inflammation, such as interleukin-1 and TNF-alpha. Why?
· There are many research showing that sex of donors has impact on many blood parameters. Have they ever done experiments with this in mind?
Author Response
First of all, on behalf of all co-authors I would like to thank you for your time and suggestions which make this manusrcipt much better. We have accepted all suggestions, and all changes in the revised version of the manuscript are marked in red.
Minor
English language and style are fine/minor spell check required.
Response: We have done the check spelling of the revised manuscipt.
I suggest to review the style of the manuscript according to the guidelines of the journal.
Response: We reviewed the style of the manuscript and, for example, separated section Results into subsections, and also we made all necessary changes according to the guidelines of the journal
All acronyms in the Abstract and Manuscript should be defined when first appear
Response: We changed the acronymes as requested.
Major
There is an increasing evidence that T2DM is associated with oxidative stress. Oxidative stress is critically involved in diabetes pathogenesis and hyperglycemia is the principal factor in the early stages of development. In Blood, erythrocytes are mainly exposed to glucose and other molecules present in the blood (i.e. oxidant compounds), and they have been widely investigated for their important role in different physiological conditions due to the fact of their metabolism and sensitivity to oxidative stress. Thus, the authors should add information about the link between oxidative stress increase and red blood cells. I suggest some recent papers (10.3390/antiox9050365) (10.1371/journal.pone.0235335), in order to provide a general more knowledge.
Response: We agree with this suggestion, and we have inserted a paragraph with suggested references at page 13 (marked in red): „There is an increasing evidence that oxidative stress is involved in pathogenesis of T2DM. On the other hand, hyperglycemia, as one of the hall-marks of diabetes, influences early stages of T2DM development. For example, erythrocytes from blood are exposed to an increased glucose which can affect integrity of their cell membranes, production of ROS, changes in hexosomonophospate shunt, glutathione cycle and antioxidative defence. Increased glucose level also influences hemoglobin modification, reflecting on Bp3 function [25, 26].“
Aim should be explain better
Response: On page 3 of the manuscript we added the following text (marked in red): “Based on this, we made a hypothesis that PBMCs in prediabetic and diabetic persons may be more susceptible to creation of DNA damage and oxidative stress following the exposure to T3, in comparison to the PBMCs of normal persons. Thus, we decided to investigate….”
In Figure 1, total comet score, * p <0.05; ** p <0.01; *** p <0.001 (Bonferroni test) -comparisons with the negative control (0 point)- is showed. In the caption, a double comparison is missing. However, the author indicate the following test Bonferroni test. Statistical analyzes should be explain better. Maybe by Dunnett test.
Response: Bonferoni test was mentioned by mistake, all relevant analysis were done with Dunnett test instead. We corrected this in the manuscript (marked in red).
The authors investigated about the assessment of oxidative stress in peripheral blood mononuclear cells. However, ROS levels are missing. Why? It could be a direct evidence when oxidative stress is produced in excess.
Response: At page 14 we inserted the following text: “Despite the results obtained in this study, one of the limitations is that we did not analyse concentrations of ROS after the exposure of PBMCs to T3. Since the halflife of ROS is very short, we prefered to analyse more stable parameters - antioxidative enzymes and damage of the cell membranes instead.”
Oxidative stress is often associated to inflammation. No evidence are showed about the link between ROS production and targets of inflammation, such as interleukin-1 and TNF-alpha. Why?
Response: We added the following text (pages 14-15), with two new references (35,36): “Moreover, ROS can be closely linked with progression of infalmmatory disorders and tissue injury. Although deleterious to cells at high enough concentrations, at low concentrations ROS can serve as signalling molecules capable to regulate cell growth, differentiation, cell senescence and apoptosis. The chronic or prolonged ROS production have an important role in the progression of inflammatory disorders. ROS produced by mitochondrions are implicated in chronic inflammation, cancer progression, diabetes mellitus, atherosclerosis, and also contribute to LPS-mediated production of pro-inflammatory cytokines IL-1β, IL-6, and TNF-α [35]. These data point to the relationships of ROS production, inflammation and damage of biomolecules. In our study all patients were overweght (25<BMI<30), or obese (BMI>30). Increased waist circumference is characterised by inflammation in lipid tissue and decerased antioxidative defence, leading to increase of ROS production, especially in T2DM patients [36].”
There are many research showing that sex of donors has impact on many blood parameters. Have they ever done experiments with this in mind?
Response: Sex of donors can influence many different results, but at the Clinic for Endocrinology, Diabetes and Metabolic Diseases most of the patients were women, they also take more care about regular controls, they are also try to prevent chronic diseases more than men, so that was the reason why we chose women for this study.
Round 2
Reviewer 2 Report
Accept in present form